# Changes in Suicide Rate and Characteristics According to Age of Suicide Attempters before and after COVID-19

**DOI:** 10.3390/children9020151

**Published:** 2022-01-25

**Authors:** Min-Jung Kim, So-Hyun Paek, Jae-Hyun Kwon, Soo-Hyun Park, Hyun-Jung Chung, Young-Hoon Byun

**Affiliations:** CHA Bundang Medical Center, Department of Emergency Medicine, CHA University, Seongnam 13496, Korea; mjtear@naver.com (M.-J.K.); rirengarnat@chamc.co.kr (J.-H.K.); suas11@chamc.co.kr (S.-H.P.); sobluebut87@chamc.co.kr (H.-J.C.); byunyoun84@chamc.co.kr (Y.-H.B.)

**Keywords:** suicide attempt, COVID-19, adolescent, adult, elderly

## Abstract

This study aims to identify age-related suicide-related factors and changes in suicide rate before and after the onset of the COVID-19 pandemic. Methods: From 2018 to 2020, the patients who presented to the ED of a university hospital with a suicide attempt were classified into adolescents (≤18 years), adults (19–65 years), and elderly (>65 years), and the visits were grouped into before and after COVID-19. Results: There were 853 visits before and 388 visits after COVID-19, and the results showed that the number of adolescent and adult suicide patients increased immediately after the pandemic, but the overall trend did not show a significant difference from before the pandemic. In the adolescents, the ratio of male patients increased, interpersonal and school-related motivations decreased, the poisoning and cutting methods of suicide were more common, and hospitalization admissions increased. Among the elderly, the ratio of female patients increased, the number of single patients and patients without previous psychiatric problems increased, the motives for physical illness and death of people around increased, the falling and hanging methods of suicide were more common, and hospitalization admissions and deaths increased. Conclusion: The impact of COVID-19 on suicide rates and suicide-related factors varies by age group. This finding requires different approaches and methods to suicide prevention based on age.

## 1. Introduction

The COVID-19 pandemic broke out in China at the end of 2019 and rapidly spread throughout the world [1]. In South Korea, the first case was confirmed in January 2020 [2], and the number of cumulative confirmed cases reached 340,000 in October of 2021 with over 1000 daily confirmed cases. Several studies have shown that the COVID-19 pandemic poses a threat to mental health, including anxiety, depression, and even PTSD-like reactions [3,4,5] and increased suicide rates [6,7,8,9,10,11,12,13,14,15,16].

According to previous studies, suicide rates rose during the outbreak of a pandemic such as the Spanish flu and SARS, and mental health threats continued for about three years even after the end of the outbreak [17,18]. The COVID-19 pandemic, which has been ongoing for nearly two years worldwide, is considered an unprecedented disastrous situation causing significant mental, emotional, and psychological pain among many people.

Suicide occurs in every age group, and according to the data from the 2020 Statistics Korea [19], suicide is the number one cause of death in South Korea for those in their 10s–30s, the second leading cause among those in their 40s–50s, and the fourth leading cause among those in their 60s. Suicide rates vary across adolescents, adults, and the elderly [20], and several recent studies have also shown that the clinical characteristics and risk factors of suicide also vary by age [21,22,23].

However, few studies have examined the effects of COVID-19 on age-specific suicide rates and characteristics. Additionally, since patients who have attempted suicide mainly visit the emergency department of a hospital, analyzing them can help provide clinically important information.

Therefore, this study aims to analyze whether there are changes in the suicide rate by age and the clinical characteristics before and after the COVID-19 pandemic.

## 2. Materials and Methods

### 2.1. Research Design and Setting

First, in this study a retrospective review of the electronic medical records was conducted and it was identified that the subjects were patients who visited the emergency department of a university hospital for suicide attempts between 1 January 2018 and 31 December 2020. This hospital has an average of 55,000 patients visiting the Adult Emergency Department per year. The Pediatric Emergency Center is one of the five designated centers in South Korea with 25,000 visits per year and treats all patients under the age of 18.

The infectious disease risk level was upgraded to its highest level “Red” after the first confirmed case of COVID-19 (23 February 2020) [24] in South Korea as the outbreak was recognized as a pandemic.

This study was approved by the hospital’s institutional review board (CHAMC 2021-10-016).

### 2.2. Selection of Participants

Patients who visited the Emergency Department with attempted suicide and/or self-harm for the purpose of suicide were enrolled. Self-harm patients who did not visit for suicide purposes and patients who visited for suicidal thoughts without attempts were excluded (based on the patient’s statement). The patients were classified into adolescents 18 years old or younger, adults between 19 and 65 years old, and the elderly over 65 years old, and the changes before and after COVID-19 in each group were compared.

### 2.3. Data Collection

First, the medical records and data collected through the standard patient record form were reviewed, and the number of suicide attempts before and after COVID-19 was analyzed (Figure 1). Second, the differences in characteristics related to suicide attempts were analyzed by dividing them into before and after COVID-19.

Baseline characteristics were analyzed such as gender, marital status, underlying disease, and insurance type. Suicide attempt-related characteristics were alcohol consumption at the time of suicide, current suicidal thoughts, suicide plans, joint suicide, motive for suicide attempt, previous suicide attempt, and psychiatric history. Finally, this study analyzed the methods and results of emergency department discharge.

### 2.4. Data Analysis

Data were analyzed using SAS statistical software (SAS system for Windows, ver. 9.4; SAS Institute, Cary, NC, USA). Categorical variables were expressed as frequency and percentage, and Pearson’s chi square test and Fisher’s exact test were performed. A *p*-value of <0.05 was considered statistically significant.

## 3. Results

A total of 1292 patients visited the emergency department for suicide or self-harm from 1 January 2018 to 31 December 2020. A total of 51 patients were excluded from the study as self-harm patients without suicidal thoughts or patients only with suicidal thoughts, and a total of 1241 patients were finally enrolled in this study (Figure 1). Of the 1241 enrolled patients, 853 patients were before COVID-19 and 388 patients were after COVID-19.

There was a total of 119 adolescents under the age of 19, 1010 adults aged 19–65, and 112 aged 66 or older. There was no statistically significant difference in the number of patients before and after COVID-19 (Figure 1).

When adolescents under 19 years of age were divided into three groups (Children (<13 years), early adolescents (13–16 years), and late adolescents (17–18 years)), there was a statistically significant increase in suicide attempts by late adolescents after COVID-19 (late adolescent (17–18 years) 50.7% vs. 66.7% (*p* = 0.013)) (Figure 1).

The trend of the number of suicide attempts by month from January 2018 is shown in Figure 2. Given that the period after 23 February 2020 was upgraded to red in the infectious disease crisis level, the number of suicide attempts increased immediately after the pandemic (Figure 2A). As for the number of suicide patients by age, the number of suicide attempts among adolescents and adults increased, and the number of suicide attempts among the elderly decreased immediately after the outbreak of the pandemic (Figure 2D).

### 3.1. Baseline Characteristics

After COVID-19, the ratio of male suicide-attempt patients in adolescents (16.9% vs. 30.9%, *p* < 0.001) and the ratio of female elderly patients increased (43.8% vs. 56.3%, *p* < 0.001) (Table 1).

In the adult and elderly groups, single and widowed statuses were statistically more significant after COVID-19 (single 2.5% vs. 9.4%, widowed 10% vs. 21.9% in elderly group, *p* < 0.001). In the adult group, there were more suicide attempts in healthy patients after COVID-19 (71.6 vs. 81.2%, *p* < 0.001), and in the elderly group, suicide attempts by patients with underlying diseases were statistically significantly higher (Table 1). As for the insurance type, the number of suicide attempts by uninsured patients in the elderly group was higher than the number before COVID-19 (7.5% vs. 21.9%, *p* = 0.010) (Table 1).

### 3.2. Suicide Attempt Related Characteristics

After COVID-19, the impulsive suicide ratio increased statistically significantly in all age groups compared to before (*p* < 0.001, Table 2). Regarding this, the number of suicide attempts among elderly patients without prior psychiatric problems increased after COVID-19 (55% vs. 65.6%, *p* = 0.008). Rather, adolescent suicide attempts were more common in patients with psychiatric problems (50.7 vs. 69.1%, *p* = 0.008) and those with a history of suicide attempts (53.2 vs. 71.4, *p* < 0.001) after COVID-19.

Alcohol consumption at the time of suicide increased among adult patients after COVID-19 (271 (38.9) vs. 137 (43.6) *p* < 0.001), which was statistically significant. (Table 2).

Motives for suicide attempts were more related to mental problems after COVID-19 for all ages. In adolescents, interpersonal relationships and job-related motivation (school) decreased, while in the elderly, the motives for physical illness or loss of people around them had a statistically significant increase (*p* < 0.001) (Table 2).

As for suicide attempt methods before and after COVID-19 by age, poisoning and cutting increased in adolescents (poisoning 50.6% vs. 52.4%, cutting 31.2% vs. 40.5%, *p* < 0.001), while falling and hanging increased more in the elderly (falling 11.2% vs. 12.5%, hanging 12.5% vs. 21.9%, *p* < 0.001) (Table 2).

## 4. Discussion

There are several studies on the suicide rate during COVID-19 [6,7,8,9,10,11,12,13,14,15,25,26,27,28], and this study is the first to compare suicide rates and characteristics according to age groups before and after COVID-19.

The results of this study could confirm the increase in the number of suicide patients after the pandemic as in the previous study [9,26,29]. When analyzed by age group, the suicide rate after the COVID-19 pandemic increased the proportion of adolescent males, and adult and elderly females (Table 1).

According to previous studies [13,14,30], it is known that the suicide rate among women has increased after the COVID-19 pandemic. In this study, the suicide rate among women increased by 3.8% (64.3–65.5%) after the pandemic. In terms of age group, suicide rate decreased to 14.7% (83.1–69.1%) in those under the age of 19 but increased by 1.26% (64.7–65.9%) in those between the age of 19–65 and especially increased by 12.5% (43.8–56.3%) in the elderly aged over 65 (Table 1).

Adult patients over 19 years of age could see an increase in the suicide rate of unmarried and widowed patients compared to before the COVID-19 pandemic. This appears to be a risk factor for suicide attempts if alone, which is consistent with several studies showing that loneliness is associated with increased suicidal thoughts [30,31].

After the COVID-19 pandemic, more patients attempted suicide while drinking, especially among patients aged 19–65 years. In addition, impulsive suicide increased in all age groups. This can be interpreted according to the study results that suggest drinking and drinking-related problems (including self-harm and suicide attempts) are on the rise after COVID-19 [32], impulsivity increased during COVID-19 [33,34], and that drinking can also increase impulsivity and suicide attempts [34]. These results show the characteristics of the pandemic era, which seem to facilitate impulsive suicide attempts due to the social atmosphere during the COVID-19 period.

Furthermore, the suicide rate increased among patients with a history of psychiatric history, but it increased in adolescents and decreased in other age groups. Adolescent suicide is known to be closely related to mental disorders and previous suicide attempts [35]. One study shows that people with a psychiatric history have more impulsive and suicidal thoughts than healthy people after COVID-19 [36], which is consistent with the results of this study. However, another previous study [37] shows that psychiatric problems were risk factors for suicide in the elderly, but the suicide attempt rate of patients without psychiatric problems was higher in this study.

This can be considered a change due to COVID-19, since older people may be particularly more vulnerable to suicide through a greater risk of disconnection from society, physical distancing, and the loss of everyday social opportunities in an epidemic environment with prevalent social lockouts [25]. This suggests that an environment can increase suicidal thoughts even for patients without a psychiatric history.

After the COVID-19 pandemic, the motive for suicide attempt has become most related to psychiatric problems for all age groups. Compared to before COVID-19, interpersonal (11.9%) and work-related (4.7%) motivation decreased in adolescents; work-related (2.9%) decreased in adults; and physical illness (25%) and the death of others or serious illness (9.4%) became more common in the elderly. Prior studies have confirmed that interpersonal and school-related problems are more likely to motivate adolescents than other age groups to commit suicide [37,38]. This is a positive result from a decrease in academic and peer conflicts due to school closures and online classes held after the COVID-19 outbreak, which is consistent with other research results [27,39]. Moreover, physical illness and death of people around the elderly can be the main causes of suicide compared to other age groups [40,41], and other studies confirm that there are more physical problems in the motives of suicide among the elderly after COVID-19 [42].

After COVID-19, poisoning (52.4%) and cutting (40.5%) increased and general hospitalization increased (33.3%) in adolescents. This is consistent with the results of other study that post-COVID-19 poisoning and the severity increased [43]. Next, the elderly used falling (12.5%) and hanging (21.9%) methods of suicide more, and as a result, ICU admissions (15.6%) and deaths (31.3%) increased. One study shows that the elderly use a harder method compared to other age groups [44], while another study indicates that the severity of suicide patients after COVID-19 increased [43]. In this study, harder methods such as falling and hanging were used more after COVID-19 even in the same elderly age group, which suggests the increase in deaths from this. This confirms that COVID-19 has contributed to the fatality of suicide methods.

As described above, the COVID-19 pandemic has a significant impact on suicide attempts across all age groups. The COVID-19 pandemic has not yet shown any signs of slowing down, so we need to think about how to manage suicide risk during the pandemic. Based on the results of this study, there are approximately two methods that can be suggested for suicide risk management:As the risk of suicide attempts among people with pre-existing mental disorders has increased, policy help is needed to actively manage mental health by increasing the use of telemedicine and other digital means [45].Since social isolation, a strategy for mitigating the risk of spreading the virus, could increase the risk of suicidal thoughts, it would be helpful to increase social welfare services for the socioeconomically underprivileged and the elderly living alone [46].

This study is significant in that it compared the suicide attempt rate and suicide characteristics of all age groups before and after the COVID-19 pandemic, but there are also some limitations as follows:As a retrospective study, this study may have selection bias and confounding variables.It is difficult to generalize the results of this study since it targeted patients who visited one university hospital. The results may be generalized by using NEDIS, which is a database for patients visiting emergency departments across the country.This study did not objectively evaluate the severity of suicide attempts. Further studies are needed to objectively evaluate the severity through known scales such as Columbia-Suicide Severity rating Scale (C-SSRS) [47] and The scale for assessment of lethality of suicide attempt (SALSA) [48] in order to find factors related to the severity.Since the COVID-19 pandemic is still ongoing, it is impossible to conclude the impact of COVID-19 on suicide based on the first year of the outbreak. Accordingly, it may be necessary to conduct a follow-up study on suicide attempts after COVID-19 is over depending on the progress of the pandemic.This study tried to report the factors related to suicide attempts before and after COVID-19 by age, but due to the short period and lack of N numbers, the multivariate model could not find any significant related factors. In the future, we plan to conduct multivariate analysis by collecting additional data.

## 5. Conclusions

This study showed that the impact of COVID-19 differs across age groups in suicide rates and factors associated with suicide. Due to the social atmosphere caused by COVID-19, there are many impulsive suicides in all age groups, and the main cause is the motive for attempts by mental disorders. Among adolescents, the number of hospitalized patients due to poisoning or self-harm increased, and as the number of falling or hanging cases in the elderly increased, the number of hospitalizations and deaths in the intensive care unit also increased. The results of the study show that there should be different approaches to prevent suicide among patients by age, and that it is necessary that the prevention be tailored to the characteristics of each age group.

## Figures and Tables

**Figure 1 children-09-00151-f001:**
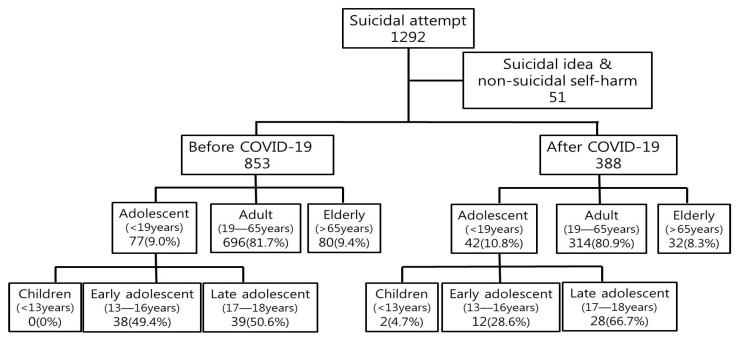
Patient flow chart. After applying the exclusion criteria, the data was divided into before and after COVID-19 and analyzed by age.

**Figure 2 children-09-00151-f002:**
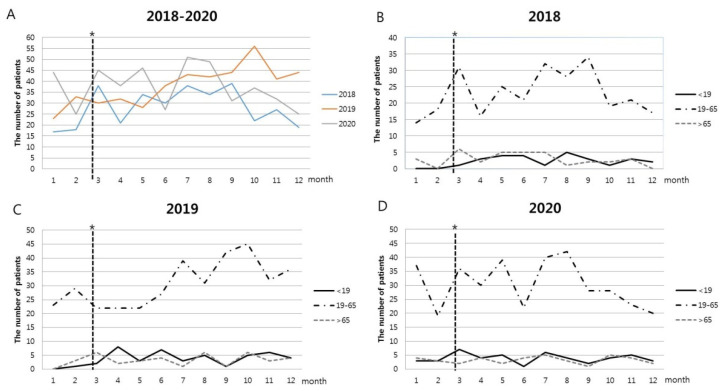
Trends in the number of suicide attempts by year. (**A**) Number of suicide attempts by year for all ages, (**B**) Number of suicide attempts by age in 2018, (**C**) Number of suicide attempts by age in 2019, (**D**) Number of suicide attempts by age in 2020. * The dotted line means that on 23 February 2020, the infectious disease crisis level was raised to red.

**Table 1 children-09-00151-t001:** Baseline characteristics before and after COVID-19.

		Before COVID-19			After COVID-19			
		Adolescent (<19 Years) (N = 77)	Adult (19–65 Years) (N = 696)	Elderly (>65 Years) (N = 80)	Adolescent (<19 Years) (N = 42)	Adult (19–65 Years) (N = 314)	Elderly (>65 Years) (N = 32)	*p*-Value
Gender, N (%)	Male	13 (16.9)	246 (35.3)	45 (56.2)	13 (30.9)	107 (34.1)	14 (43.8)	<0.001 †
	Female	64 (83.1)	450 (64.7)	35 (43.8)	29 (69.1)	207 (65.9)	18 (56.3)	
Marital status, N (%)	Single	77 (100.0)	373 (53.6)	2 (2.5)	42 (100.0)	192 (61.1)	3 (9.4)	<0.001 ††
	Married	0 (0.0)	256 (36.8)	70 (87.5)	0 (0.0)	93 (29.6)	22 (68.7)	
	Partner	0 (0.0)	4 (0.6)	0 (0.0)	0 (0.0)	4 (1.3)	0 (0.0)	
	Separated	0 (0.0)	9 (1.3)	0 (0.0)	0 (0.0)	4 (1.3)	0 (0.0)	
	Divorced	0 (0.0)	45 (6.5)	0 (0.0)	0 (0.0)	14 (4.5)	0 (0.0)	
	Widowed	0 (0.0)	9 (1.3)	8 (10.0)	0 (0.0)	7 (2.2)	7 (21.9)	
Underlying disease, N (%)	Healthy	75 (97.4)	498 (71.6)	24 (30.0)	40 (95.2)	255 (81.2)	5 (15.6)	<0.001 ††
	Recent acute illness	0 (0.0)	3 (0.4)	2 (2.5)	0 (0.0)	5 (1.6)	0 (0.0)	
	Chronic disease, not interfering with life.	1 (1.3)	124 (17.8)	31 (38.8)	2 (4.8)	32 (10.2)	13 (40.6)	
	Chronic disease, interfering with life.	1 (1.3)	71 (10.2)	23 (28.7)	0 (0.0)	22 (7.0)	14 (43.8)	
Insurance type, N (%)	NHI ^1^	66 (85.7)	570 (81.9)	66 (82.5)	37 (88.1)	280 (89.2)	24 (75.0)	0.010 ††
	Medical care 1	3 (3.9)	59 (8.5)	8 (10.0)	1 (2.5)	16 (5.1)	1 (3.1)	
	Medical care 2	2 (2.6)	22 (3.1)	0 (0.0)	2 (4.7)	5 (1.6)	0 (0.0)	
	No insurance	6 (7.8)	45 (6.5)	6 (7.5)	2 (4.7)	13 (4.1)	7 (21.9)	

NHI ^1^: National Health Insurance. †: Chi-square test, ††: Fisher’s exact test.

**Table 2 children-09-00151-t002:** Suicide attempt related characteristics.

		Before COVID-19			After COVID-19			
		Adolescent (<19 Years) (N = 77)	Adult (19–65 Years) (N = 696)	Elderly (>65 Years) (N = 80)	Adolescent (<19 Years) (N = 42)	Adult (19–65 Years) (N = 314)	Elderly (> 65 years) (N = 32)	*p*-Value
Alcohol consumption at the time of suicide, N (%)	Yes	11 (14.3)	271 (38.9)	15 (18.7)	6 (14.3)	137 (43.6)	4 (12.5)	<0.001 †
	No	66 (85.7)	425 (61.1)	65 (81.3)	36 (85.7)	177 (56.4)	28 (87.5)	
Current suicidal thought, N (%)	Yes	29 (37.7)	237 (34.1)	25 (31.3)	19 (45.2)	94 (29.9)	6 (18.7)	0.139 †
	No	48 (62.3)	459 (65.9)	55 (68.7)	23 (54.8)	220 (70.1)	26 (81.3)	
Suicide plan, N (%)	Yes	6 (7.8)	60 (8.6)	6 (7.5)	2 (4.8)	3 (1.0)	1 (3.1)	<0.001 ††
	No	71 (92.2)	636 (91.4)	74 (92.5)	40 (95.2)	311 (99.0)	31 (96.9)	
Joint suicide, N (%)	Yes	0 (0.0)	9 (1.3)	7 (8.7)	1 (2.4)	3 (1.0)	0 (0.00)	<0.001 ††
	No	77 (100.0)	687 (98.7)	73 (91.3)	41 (97.6)	311 (99.0)	32 (100.00)	
Motive for suicide attempt, N (%)	Psychiatric	13 (16.9)	123 (17.7)	9 (11.3)	27 (64.3)	112 (35.7)	8 (25.0)	<0.001 ††
	Inter-personal	18 (23.4)	88 (12.6)	2 (2.5)	5 (11.9)	58 (18.5)	3 (9.4)	
	Work-related	10 (13.0)	23 (3.3)	1 (1.2)	2 (4.7)	9 (2.9)	0 (0.0)	
	Economic	0 (0.0)	38 (5.5)	0 (0.0)	0 (0.0)	14 (4.5)	2 (6.2)	
	Physical illness	0 (0.0)	22 (3.2)	14 (17.5)	1 (2.4)	8 (2.5)	8 (25.0)	
	Death of others or serious illness	0 (0.0)	7 (1.0)	1 (1.2)	0 (0.0)	3 (0.9)	3 (9.4)	
	Fight	9 (11.7)	111 (15.9)	9 (11.3)	6 (14.3)	29 (9.2)	1 (3.1)	
	Etc.	27 (35.0)	284 (40.8)	44 (55.0)	1 (2.4)	81 (25.8)	7 (21.9)	
Previous suicide attempt, N (%)	Yes	41 (53.2)	220 (31.6)	6 (7.5)	30 (71.4)	105 (33.4)	3 (9.4)	<0.001 †
	No	36 (46.8)	476 (68.4)	74 (92.5)	12 (28.6)	209 (66.6)	29 (90.6)	
Psychiatric Hx., N (%)	Yes	39 (50.7)	407 (58.5)	36 (45.0)	29 (69.1)	175 (55.7)	11 (34.4)	0.008 †
	No	38 (49.3)	289 (41.5)	44 (55.0)	13 (30.9)	139 (44.3)	21 (65.6)	0.008 †
Suicide attempt methods, N (%)	Poisoning	39 (50.6)	412 (59.2)	51 (63.8)	22 (52.4)	189 (60.2)	20 (62.5)	<0.001 ††
	Cutting	24 (31.2)	171 (24.6)	10 (12.5)	17 (40.5)	78 (24.8)	0 (0.0)	
	Falling	9 (11.7)	32 (4.6)	9 (11.2)	2 (4.7)	16 (5.1)	4 (12.5)	
	Hanging	3 (3.9)	51 (7.3)	10 (12.5)	1 (2.4)	27 (8.6)	7 (21.9)	
	Other method	2 (2.6)	30 (4.3)	0 (0.0)	0 (0.00)	4 (1.3)	1 (3.1)	
Results of ED discharge, N (%)	Discharge	56 (72.7)	518 (74.5)	34 (42.5)	26 (61.9)	246 (78.3)	16 (50.0)	<0.001 ††
	Transfer	0 (0.0)	5 (0.7)	1 (1.2)	2 (4.8)	9 (2.9)	0 (0.0)	
	General hospitalization	13 (16.9)	85 (12.2)	16 (20.0)	14 (33.3)	35 (11.2)	1 (3.1)	
	ICU ^1^ admissions	5 (6.5)	44 (6.3)	10 (12.5)	0 (0.0)	8 (2.5)	5 (15.6)	
	Deaths	3 (3.9)	44 (6.3)	19 (23.8)	0 (0.0)	16 (5.1)	10 (31.3)	

ICU ^1^: Intensive Care Unit. †: Chi-square test, ††: Fisher’s exact test

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
