# Peer review of "Changes in Suicide Rate and Characteristics According to Age of Suicide Attempters before and after COVID-19"

_children, 2022, doi:10.3390/children9020151_

Round 1

Reviewer 1 Report

This is an interesting paper aiming to evaluate the suicide trends in a sample of the general population in China, using accessing data to emergency departments in the period 2018-2020.

The topic is timely and interesting, I have some minor suggestions to further improve the paper:

  • In the introduction, authors should discuss on the role of the COVID-19 pandemic as a traumatic stressor, impacting on mental health. Several papers have been published on this topic, such as Bridgland VME, Moeck EK, Green DM, Swain TL, Nayda DM, Matson LA, Hutchison NP, Takarangi MKT. Why the COVID-19 pandemic is a traumatic stressor. PLoS One. 2021 Jan 11;16(1):e0240146; Clemente-Suárez VJ, Martínez-González MB, Benitez-Agudelo JC, Navarro-Jiménez E, Beltran-Velasco AI, Ruisoto P, Diaz Arroyo E, Laborde-Cárdenas CC, Tornero-Aguilera JF. The Impact of the COVID-19 Pandemic on Mental Disorders. A Critical Review. Int J Environ Res Public Health. 2021;18(19):10041; Marazziti D, Stahl SM. The relevance of COVID-19 pandemic to psychiatry. World Psychiatry. 2020 Jun;19(2):261.
  • Among study's limitations, authors should acknowledge that they did not evaluate the severity of suicidal attempts/suicidal ideation. 
  • on p. 3, line 101, there is a typo (I think it should be "2020" instead of "2000").
  • In the Discussion, authors should provide some practical suggestions on how to manage (and hopefully reduce) suicide risk during the pandemic. You should quote the paper by Wasserman D, Iosue M, Wuestefeld A, Carli V. Adaptation of evidence-based suicide prevention strategies during and after the COVID-19 pandemic. World Psychiatry. 2020 Oct;19(3):294-306; McIntyre RS, Lee Y. Preventing suicide in the context of the COVID-19 pandemic. World Psychiatry. 2020 Jun;19(2):250-251.

Reviewer 2 Report

This is a well written and important paper. The results unfortunately show and support the tendency we find worldwide: “late adolescence men are especially vulnerable, and more and more  elderly with physical illness are at risk for suicide” due to Covid and its measures against it.

However I have some (1)  concerns.

The methodology and statistics are quite simple. The time frames are different which makes that we a different N (power issues )and it clearly shows that the population on the ED changes after Covid especially by adolescent. In the discussion the authors tend to contextualize their results in terms of relations between predictors/risk factors and suicide behavior.  However, It is not clear why they did not test those relations  in a multivariate model with for example pre and post Covid as an interaction term. I can imagine that the authors want to collect more data for this, but it is certainly a question that will  pop up by the reader. Probably multivariate analysis will learn us more and will attract more attention in the suicide research community. If this is not possible the authors should explain  and discuss why they are choosing for a relative simple analysis and potential bias because of that. Also needed  in the discussion a bridge to such multivariate models in general
